# A Cost-Effective Reusable Tissue Mimicking Phantom for High Intensity Focused Ultrasonic Liver Surgery

**DOI:** 10.3390/bioengineering9120786

**Published:** 2022-12-09

**Authors:** Sitaramanjaneya Reddy Guntur, Seong-Chan Kim, Min-Joo Choi

**Affiliations:** 1Department of Biomedical Engineering, Vignan’s Foundation for Science, Technology, and Research, Vadlamudi, Guntur 522213, India; 2Interdisciplinary Postgraduate Program in Biomedical Engineering, Jeju National University, 102 Jejudaehak-ro, Jeju-si 63243, Jeju Special Self-Governing Province, Republic of Korea; 3Department of Medicine, College of Medicine, Jeju National University, 102 Jejudaehak-ro, Jeju-si 63243, Jeju Special Self-Governing Province, Republic of Korea

**Keywords:** ultrasonic tissue-mimicking phantom, high intensity focused ultrasound (HIFU), polyacrylamide polysaccharide hydrogel (PASG), nonionic surfactant, visualization, thermal lesion

## Abstract

A polyacrylamide polysaccharide hydrogel (PASG) containing a nonionic surfactant of the polyoxyethylene nonylphenyl ethers series (NP14) has been adapted to the fabrication of a reusable cost-effective ultrasonic tissue-mimicking phantom for real-time visualization of the thermal lesions by high intensity focused ultrasound (HIFU) irradiation. The constructed NP14 (40% in *w*/*v*) PASG is optically transparent at room temperatures, and it turns out to be opaque white as heated over the clouding points of about 55 °C and returns to its original transparent state after cooling. The acoustic property of the proposed phantom is similar to those of human liver tissues, which includes the acoustic impedance of 1.68 Mrayls, the speed of sound of 1595 ± 5 m/s, the attenuation coefficient of 0.52 ± 0.05 dB cm^−1^ (at 1 MHz), the backscatter coefficient of 0.21 ± 0.09 × 10^−3^ sr^−1^ cm^−1^ (at 1 MHz), and the nonlinear parameter B/A of 6.4 ± 0.2. The NP14-PASG was tested to assess the characteristic information (sizes, shapes, and locations) of the thermal lesions visualized when exposed to typical HIFU fields (1.1 MHz, focal pressure up to 20.1 MPa, focal intensity 4075 W/cm^2^). The proposed NP14-PASG is expected to replace the existing costly BSA-PASG used for more effective testing of the performance of therapeutic ultrasonic devices based on thermal mechanisms.

## 1. Introduction

Noninvasive thermal ablative modalities such as high-intensity focused ultrasound (HIFU) have drawn increasing attention, particularly for localized tumor ablation [1,2]. To minimize in vivo and clinical tests, phantom studies are often employed, visualizing the lesions induced by ablation therapies. For instance, in HIFU surgery making use of ultrasonic thermal effects, an ultrasonic tissue-mimicking phantom (UTMP) that is temperature sensitive and optically transparent has been used to visualize the thermal lesion formation.

One of the popular phantoms that have been employed for testing the HIFU-induced thermal lesions may be the polyacrylamide hydrogel (PAG) with protein (e.g., bovine serum albumin (BSA) or egg white) as a temperature-sensitive agent. The phantom is transparent and turns out to be white opaque when the protein is denaturized by ultrasonic thermal excitation exceeding a threshold temperature of around 70 °C [3,4]. This allows us to visualize the thermal lesion formed at temperatures above the threshold for protein denaturation [3]. The thermal lesion visualized in the BSA-PAG may be regarded as the thermal necrosis of a target (e.g., cancerous) tissue.

The thermal lesion can appear in the BSA-PAG when the temperature is raised above a fixed threshold temperature level. What is visualized in the BSA-PAG is the protein denaturation which may be useful in specific ultrasonic thermotherapy. In clinical studies, the effectively treated areas (e.g., thermal necrosis of cancerous tissue) may not always coincide with those with the boundary at the protein denaturation temperature [5]. In addition, the denaturation of BSA is an irreversible process through the fixed temperature threshold of about 70 °C. Once the thermal lesion is formed in the PAG, it is permanent, and the expensive phantom may not be used again. In order to overcome those shortcomings, a novel approach was suggested in our previous study [6], where non-ionic surfactant (NiS) was adopted to replace BSA as a novel temperature-sensitive indicator.

NiS exhibits hydrophobic segregation when the temperature exceeds a specific threshold level known as the “clouding point” [7,8], causing it to become opaque white. The threshold temperature above which a thermal lesion is formed in the PAG is controlled by selecting a suitable NiS with a different clouding point and by adjusting the concentration of NiS. The PAG containing NiS (e.g., OA10, TDA10, OTA8, etc.), was designed to be transparent at room temperatures. The NiS-PAG gets segregated to be opaque, when it is ultra-sonically heated over the threshold temperature, for example, 53 °C to the NiS of TDA10. This change is reversible upon cooling, allowing us to reuse the phantom.

The above-mentioned reusable UTMPs still have drawbacks. For instance, they have ultrasonic attenuation coefficients even lower than tissues and do not scatter ultrasonic waves. The acoustic property can give a significant effect on the location and size of the thermal lesions formed in the phantoms by HIFU. An improved phantom was suggested in our previous work [9], based on BSA-PAG to which polysaccharide and ultrasonic scatterers were added to increase ultrasonic attenuation. The acoustic property of the improved BSA-PAG was shown to be similar to that of liver tissue. The polyacrylamide polysaccharide hydrogel (PASG) phantom hereafter is named the “BSA-PASG” instead of an “improved BSA-PAG”.

Another problem would be that the threshold temperatures of the existing reusable phantoms are far lower than that for protein denaturalization (~70 °C). For instance, they are 48 °C, 53 °C and 57 °C for the PAGs containing NiS OA10, TDA10 and OTA8, respectively. N-isopropyl acrylamide and polyNIPAM copolymers, reported as other reusable phantoms, are 42 °C and 52 °C [10,11]. This indicates that the existing reusable phantoms in their present forms may not be suitable for routine clinical tests in HIFU tumor surgery.

The study was to propose a reusable cost-effective UTMP made of a polyacrylamide polysaccharide hydrogel containing NP14 of an NiS of the polyoxyethylene nonylphenyl ethers series, as a temperature-sensitive agent. This allows the optical visualization of the thermal lesion produced by the HIFU used for liver tumor ablation and replaces the existing costly BSA-PASG. The NP14 was selected to utilize not only its clouding process being equivalent to protein denaturation in the BSA-PASG for heating but also its reversible process for cooling making the proposed phantom reusable. The proposed NP14-PASG was designed to be acoustically similar to the biological scattering of liver tissues. The onset of the NiS clouding in the NP14-PASG represents the temperature above which the liver tumor is thermally destroyed. Experiments were performed to visualize the thermal lesions formed in the constructed NP14-PASG and to verify the thermal tissue destruction by HIFU for liver tumor ablation.

## 2. Methods

### 2.1. Preparation of NP14-PASG

NP14-PASG UTMP employs PASG that contains NP14 of NiS nonylphenol ethoxylate series (IC Chemical Ltd., Busan, Republic of Korea) as a thermo-sensitive indicator. Of many nonylphenol ethoxylate series, the NP14 was chosen to make the UTMP whose threshold temperature can visualize protein denaturation of the thermal necrosis for the liver tumor ablation. The NP14-PASG was prepared in the same way as described in our previous studies for the BSA protein phantoms [9,12], replacing BSA with NP14. Polysaccharide was included to increase the ultrasonic attenuation and stiffness of PASG, and its concentration was adjusted to achieve the acoustic parameter being similar to that of liver tissue. Table 1 shows the recipe for a volume of 50 mL of the NP14-PASG UTMP.

The fabrication process is as follows. The polysaccharide (corn syrup) (Chungjungwon, Daesang Corporation, Seoul, Republic of Korea) (20 g or 40 %(*w*/*v*)) was dissolved in 25 mL degassed distilled water before adding the 2 g of NiS NP-14 (IC Chemical Ltd., Busan, Republic of Korea) to the aqueous solution. The solution was gently stirred until the NP-14 and polysaccharide (corn syrup) were completely dissolved in distilled water. The 8.75 mL of an aqueous solution of 40 %(*w*/*v*) acrylamide with a 19:1 ratio of acrylamide: bis monomer cross-link ratio (A9926, Sigma Chemicals, St. Louis, MO, USA), and the 0.25 mL of 5 %(*v*/*v*) of 1 M TRIS buffer and the 0.15 g of 0.2 %(*w*/*v*) of sodium azide were added as a preservative. The entire solution was placed in a vacuum chamber (OV-01/02, Jeio Tech, Seoul, Republic of Korea) under a 760 mmHg vacuum for more than 60 min to remove gases. After that, the glass beads of 0.002 %(*w*/*v*) were added to the solution, which was then mixed to ensure that all glass beads were evenly distributed. Finally, 0.25 mL ammonium persulphate (APS, A7460 Sigma Chemicals, St. Louis, MO, USA) and the 0.15 mL of a polymerization agent N, N, N′, N′-tetramethyl ethylene/diamine (TEMED, T2694 Sigma Chemicals, St. Louis, MO, USA) were added to the mixture. Deionized water was added to obtain the solution be 50 mL. The final solution was immediately transferred to the container to allow polymerization to occur at room temperature (25 °C). After polymerization, the PASG was sealed in an airtight container to prevent dehydration and stored at room temperature. 

### 2.2. Measurements of Physical Properties

The acoustic, mechanical, and thermal properties of the NP14-PASG were measured at room temperature (~23 °C). A rectangular specimen (5 × 5 × 1 in cm) was used for the measurements. The acoustic properties including speed of sound, attenuation coefficient, backscatter coefficient, and B/A were measured using a broadband pulse-echo transmission technique and a standard substitution method [13,14,15]. The experimental configurations and measurements were described in our previous works [6,16,17]. The mechanical properties, such as shear and Young’s modulus, were measured while the applied force was measured with an indentor or actuator as described in Guntur and Choi et al. (2014) [9]. The thermal properties were measured using a KD-2 Pro thermal property analyzer (Decagon Devices Inc., Pullman, WA, USA), including specific heat, thermal conductivity, thermal resistivity, and diffusivity, as the details are provided in Choi et al. (2013) [17].

### 2.3. Measurements of Optical Property

The constructed NP14-PASG is optically transparent at room temperature. When temperature exceeds a threshold level, the NP14-PASG becomes opaque due to NP14 segregation. An experimental setup for measuring the temperature-dependent optical opacity is described in our previous study [6]. The changes in optical opacity of the NP14-PASG (in terms of 8-bit grayscale level) as a function of temperature were obtained from a series of optical images of the NP14-PASG obtained with a charge-coupled device (CCD) camera under the controlled illumination [6]. As temperature increases, the pixel value diffuses due to the NiS segregation. Changes in the opacity of an NP14-PASG sample in a water bath under the controlled temperature were captured throughout the heating process by using a CCD camera. For each image frame, a 20 × 20 pixel region of interest (ROI) near the thermocouple tip inserted into the PASG specimen was chosen for image processing. The mean grayscale pixel value of the ROI at a particular temperature was calculated as a measure of the phantom optical transparency at that temperature. The custom-built LabVIEW program was used to process the image frames captured during the heating process.

The observed sigmoid pattern of the optical transparency (in grayscale) against temperature may be characterized by the parameters defined in our previous study [6], including the slopes (_h_S, _c_S), the temperature ranges (_h_TR (= _h_T_s_~_h_T_e_), _c_TR (= _c_T_s_~_c_T_e_)), the optical transparency range (XR = X_max_ − X_min_ in grayscale) and the optical transparency contrast (C). The subscripts “h” and “c” stand for heating and cooling stages, respectively, while the “s” and “e” are, respectively, starting and ending during either heating or cooling process. The temperature distribution of the phantom was photographed in a grayscale image where the temperatures were mapped into the 8-bit grayscale pixel values of 0~255. Let g(T) be a function of temperature that gives a degree of optical transparency of the phantom in grayscale values at a temperature, X_max_ = g(_c_T_s_) + JND and X_min_ = g(_c_T_e_) − JND, where the JND represents just noticeable difference by naked eyes in grayscale values. The slope for clouding during heating process is given as
(1)Sh=XR−2⋅JNDThR

In the same way, the slope for clearing during cooling process can be written as
(2)Sc=XR−2⋅JNDTcR

It is noted that, as the temperature exceeds above _h_T_e_, the XR determines a maximum contrast to the background level (X_min_), indicating that the thermal lesion has been fully developed. For 8-bit grayscale images with the full range of pixel values from 0 to 255, the maximum contrast C (%) that can be achieved for the lesion is
(3)$$C=\frac{\mathrm{XR}}{255-X_{\min}}\times 100$$

With a CCD camera, the optical transparency can be detected with a sensitivity of a single grayscale difference. In the present study, JND was set to 3 in grayscales to ensure that it is larger than the fluctuating ranges of the upper and the lower stable state. Note that the grayscale difference just noticeable to the naked eye is approximately 8 [18,19].

### 2.4. Visualization of Thermal Lesions

The constructed NP14-PASG was tested with an experimental HIFU system. The experimental arrangement for visualization of the thermal lesions formed in the PASG exposed to HIFU is described in our previous studies [6,17]. The HIFU transducer was mounted in a water tank containing deionized and degassed water. The focus of the HIFU transducer was located at the center of a specimen of the PASG. The HIFU system was set to operate in a continuous wave mode at the frequency of 1.1 MHz, and the acoustic dose was controlled by exposure time and focal intensity. Thermal lesion formation process visualized in the PAG was recorded by two CCD cameras for the front and side views. The experimental system was controlled by a computer using software programmed in the Lab View environment (National Instruments, Austin, TX, USA).

## 3. Results

### 3.1. Physical Properties

The acoustic properties of the NP14-PASG were measured at room temperature (~23 °C), and the measurements were repeated five times on different PASG samples. The speed of sound was measured to be 1595 ± 5 m/s and the density was 1060 kg/m^3^, resulting in an acoustic impedance of 1.68 ± 0.03 Mrayls (Table 2 and Table 3). The measured speed of sound and acoustic impedance is close to those of human liver tissue, which are 1585 ± 10 m/s and 1.68 Mrayls, respectively [20,21].

The attenuation coefficient of the NP14-PASG was measured for the frequency range of 1 to 5 MHz. The results are plotted in Figure 1a, together with those of BSA-PASG [9], BSA-PAG [3], and liver [21,22]. The attenuation coefficient was shown to increase approximately linearly with frequency, which is similar to liver tissue. The attenuation coefficient at 1 MHz, which is a typical frequency used for HIFU surgery, was about 0.50 ± 0.05 dB/cm. This value is very close to that of the BSA-PASG [9] and liver tissue [23]. However, the increase rate of the attenuation coefficient with respect to frequency is lower than that of liver tissue, resulting in the discrepancy in attenuation coefficients between the PASG and the tissue being widen with increasing frequency. Note that the frequency-dependent attenuation coefficients were similar between the NP14-PASG and the BSA-PASG. This is because both of them were made by adding polysaccharides (40 %(*w*/*v)*) to PAG. The addition made the gel more viscous and more attenuating ultrasound. The NP14-PASG is shown to be slightly less attenuating than the BSA-PASG, and this may imply that BSA attenuates ultrasound a little more than NP14.

The backscatter coefficient is plotted in Figure 1b, as a function of frequency from 1 to 5 MHz together with those of the BSA-PASG and liver tissue. Note that the BSA-PAG was excluded here since it does not contain any ultrasonic scatters and therefore does not scatter ultrasound, unlike the others. Similar to the liver tissue, the backscattering coefficient increases nonlinearly (exponentially) with frequency. The measured backscatter coefficient was 0.21 (±0.012) × 10^−3^ cm^−1^sr^−1^ at 1 MHz, which is comparable to 0.27 (±0.017) × 10^−3^ cm^−1^sr^−1^ at 1 MHz for the normal human liver [24]. The nonlinear parameter (B/A) of the NP14-PASG was found to be 6.4 ± 0.2. This value is significantly higher than that of water 5.2 ± 0.05 [25], the BSA-PASG (5.9 ± 0.3) [9], but it is close to liver tissue (6.8 ± 0.4) [25].

The Young’s modulus of the NP14-PASG was measured to be 9.94 (±0.31) kPa comparable to 10.63 (±0.93) kPa for the BSA-PASG [9] and 13 ± 0.3 kPa for liver tissue [26]. The measured shear modulus was 4.29 (±0.39) kPa, comparable to 3.5 ± 0.55 kPa reported by Kruse et al. (2000) [27]. Table 2 and Table 3 summarize the measured acoustic and mechanical parameters, in contrast to those of BSA-PASG, and liver tissue [13,20,22,25].

**Table 2 bioengineering-09-00786-t002:** Acoustic properties of the NP14-PASG, in contrast to those of the BSA-PASG and liver tissue [3,9,13,22,25]. Measurements on the PASGs were carried out at room temperature (~23 °C) and five measurements were repeated on the different samples to obtain their mean and standard deviation. Note that the values in round brackets are the ratios of the mean values of the PASGs to those of liver tissue.

Acoustic Parameter	Units	Liver	BSA-PASG	NP14-PASG
Acoustic impedance	Mrays	1.68 ± 0.02	1.68 ± 0.03 (1.000)	1.70 ± 0.02 (1.012)
Speed of sound	m/s	1575 ± 10	1588 ± 9 (1.008)	1605 ± 8 (1.019)
Attenuation coefficient (at 1 MHz)	dB/cm	0.52 ± 0.03	0.51 ± 0.06 (0.981)	0.50 ± 0.09 (0.962)
Backscatter coefficient (at 1 MHz)	sr^−1^ cm^−1^	0.27 ± 0.017 × 10^−3^	0.22 ± 0.097 × 10^−3^ (0.815)	0.21 ± 0.09 × 10^−3^ (0.778)
B/A	-	6.8 ± 0.14	5.9 ± 0.3 (0.868)	6.4 ± 0.2 (0.941)

**Table 3 bioengineering-09-00786-t003:** Mechanical properties of the NP14-PASG, compared with those of the BSA-PASG and liver tissue [9,28,29]. Measurements were carried out at room temperature (~23 °C) and five measurements were repeated on the different samples to obtain their mean and standard deviation. Note that the values in round brackets are the ratios of the mean values of the PASG to those of liver tissue.

Mechanical Parameter	Unit	Liver	BSA-PASG	NP14-PASG
Density	kg/m^3^	1060 ± 10	1057 ± 13 (0.997)	1061 ± 10 (1.001)
Shear Modulus	kPa	3.5 ± 0.55	4.55 ± 0.33 (1.300)	4.29 ± 0.39 (1.226)
Young’s Modulus	kPa	13 ± 0.30	10.63 ± 0.93 (0.818)	9.94 ± 0.31 (0.765)

The thermal parameters of the NP14-PASG measured at room temperature (23 °C) are listed in Table 4, together with those of the BSA-PASG, and liver tissue [28,29]. The thermal properties are found to be close to those of the BSA-PASG [9]. It was shown that the NP14-PASG was able to withstand temperatures above 95 °C without melting, allowing it to be used in HIFU applications.

### 3.2. Optical Opacity against Temperature

The proposed NP14-PASG was observed to be homogeneous and optically transparent at room temperature. Figure 2 depicts the measured optical opacity when the PASG was heated to 95 °C and naturally cooled back to the initial room temperature (~25 °C), in contrast to that of BSA-PASG. The NP14-PASG begins to get white and opaque as the temperature rises to the clouding onset point (~25 °C) of the NiS. For the temperature ranges above the clouding threshold, the opacity of the PASG (in 8-bit grayscales) rapidly (almost exponentially) rises with temperature and gradually gets saturated, until it eventually remains unchanged to further continuous heating. This process is reversible, and the opacity returns to the initial transparent state when cooling. The recovery was observed to be hysterical at a temperature difference of about a 20 °C increase, as the curve was shifted to the right-hand side in Figure 2. The opacity against the temperature of BSA-PASG is similar to that of the NP14-PASG for the period of heating. However, the optical state of being opaque at temperatures higher than the threshold level of the BSA protein denaturation is permanent, so it remains unchanged even when cooling back to room temperature. In fact, the grayscale was observed to slowly decrease as temperatures were down to the initial. Note that, the grayscale levels of the proposed phantom at temperatures less than the thresholds were shown to be approximately 20 pixels lower compared with BSA-PASG. This result indicates that the NP14-PASG is more transparent at room temperature than BSA-PASG. This implies that the proposed NP14-PASG is likely to visualize the thermal lesions with a contrast higher than those in BSA-PASG.

Table 5 compares the measured values of the characteristic parameters TR and S for clouding and clearing as well as XR and C, for the reversible NP14-PASG and BSA-PASG. The denaturation onset temperature _h_T_s_ increases from 62 °C and clouding segregation from 57 °C by a constant interval of 0.5 °C for BSA-PASG and the NP14-PASG, respectively. The temperature magnitudes of BSA-PASG and the NP14-PASG were 17 °C and 10 °C, respectively. In contrast, the NP14-PASG has a larger slope of 6.93 gs/°C than BSA PAG, which has a slope of 7.95 gs/°C. In other words, the BSA-PASG process is faster than the NP14-PASG. In the reversible NP14-PASG, the slopes are larger for clouding than clearing; in other words, the process is faster for clouding than for clearing. However, the transparency ranges XR differ significantly due to X_max_ difference. The PASGs have small differences in X_min_ (with values of 17 and 10), and X_max_ (values of 190 to 203). As a result, the transparency ranges (XR) are also similar. The BSA-PASG has the largest of 173 grayscales, while the NP14-PASG has 193. In similar patterns to XR, the maximum contrast C varies between 72–79%. As defined in Equation (3), C is a function of XR and X_min_, the difference between X_min_ and XR is significant, and the parameters to determine the percentage of contrast are virtually the same.

### 3.3. Visualization of Thermal Lesions

The thermal lesions produced by HIFU irradiation were visualized in the ultrasonic tissue-mimicking phantoms. The PASG were exposed for 15 s to 1.1 MHz continuous HIFU with focal pressures of 14.2, 16.2, 18.0, and 20.1 MPa. Clear observation of the white lesions was possible as significant increases in the PSAG opacity with exposure time above the threshold temperatures. Figure 3 shows the video images of the thermal lesions formed in NP14-PAG (left two columns) in contrast to those of BSA-PASG (right two columns). The front views of the lesions are shown in the left columns, and the side views are in the right columns. Note that the HIFU transducer (not shown) was located on the left-hand side of the front view image.

The NP14-PASG shows a well-developed ellipsoidal-shaped lesion at a lower focal pressure of 14.2 MPa, which accelerates lesion growth of size and then distortion and migration towards the transducer as the acoustic pressure rises to 16.2 MPa. When the focal pressure was increased to 20.1 MPa, the lesion visualized in NP14-PAG grew in size and shape, distorted from ellipsoid to tadpole-shaped, and migrated toward the HIFU transducer. This pattern was similar to BSA-PASG, although there were some characteristic differences such as lesion shape and brightness. The slightly larger slope of clouding NP14-PAG at above 55 °C (see Table 5) was subjected to the surrounding illumination in the lesion images (Figure 3). For instance, the lesions appeared with a clear boundary in BSA-PASG compared to NP14-PAG. The lesions formed in NP14-PAG were expanded towards the transducer to tadpole in shape when the focal pressure was increased. Further, the formation of cavitation or bubbles was also produced in the vicinity of the main body of the lesion.

Similarly, Figure 4 shows the video images of the thermal lesions formed in NP14-PASG (left two columns) in contrast to those of BSA-PASG (right two columns) at the targeted focal region exposed to 3 s, 5 s, 9 s, 13 s, and 15 s continuous HIFU at a focal pressure of 16.2 MPa. The front views of the lesions are in the left columns, and the side views are in the right columns in each PAG. As the exposure time increased, the lesions grow in size preferably towards the left-hand side, namely, the HIFU transducer. For instance, in the 9th second, the lesion in the NP14-PASG was significantly enlarged and deformed into a messy tadpole with illumination, whereas such a significant deformation was also developed with a clear boundary in the BSA-PASG. As exposure time increased, the lesions became more complicated due to boiling, or bubble formation was more likely to occur at higher pressure levels above 16 MPa in both PASGs. The gradual increases in lesion sizes and shape in NP14-PASG for the same conditions of the HIFU exposure to BSA-PASG attributed that the temperature variation of NP14-PASG were similar to that of BSA-PASG (Figure 4). When compared to the BSA-PASG, the lesions formed in the NP14-PASG are relatively less clear boundaries and a similar shape.

### 3.4. Size of Thermal Lesions

The volumes of the thermal lesions visualized in the proposed phantom with respect to the acoustic dose and the exposure are depicted in Figure 5, compared with those of BSA-PASG. The mean values were obtained with three repeated measurements on different PASGs. The lesion boundary was determined by a chosen threshold level in 8-bit grayscale level 30 for NP14-PASG and 45 for BSA-PASG, at which the temperature was expected to be 62 °C (Figure 2), is regarded as the temperature threshold for thermal damage in the liver tissue. The acoustic dose represents energy density (kJ/cm^2^) and was defined by the product of focal intensity (spatial peak temporal average intensity, Ispta) and exposure time. As expected, the thermal lesion volume increases with the acoustic dose (Figure 5a) and also increases almost linearly with exposure time. The lesion sizes are very close and show no significant difference between the NP14-PASG and BSA-PASG.

## 4. Discussion

The proposed UTMP makes use of a NiS (NP14) that replaces BSA in the existing BSA-PASG, as a temperature-sensitive indicator, which is the reusable and economic version of the BSA-PASG. The NP14 has a nominal clouding point of 90 °C at which the clouding process takes place in a water NP14 solution. The clouding temperature in the gel was found to reduce and sensitive to the concentration of NP14.

A test was carried out for selecting a proper concentration with which NP14 influenced the optical transparency range (XR) but is less sensitive to the temperature range (TR). Figure 6 shows the optical opacity in grayscale values of the NP14-PASG on heating and cooling at the concentrations of NP14 from 1 to 4 %(*w*/*v*) without altering the other compositions as listed in Table 1. As the concentration was raised from 1 to 4%, XR increased because X_max_ increased while X_min_ remained constant, and the clouding temperature shift (hTs) decreased from 67 to 54. Table 6 shows the values of TR and S (of clouding) and XR and C for different concentrations of NP14. This led us to choose a concentration of 4% in the present study, with the starting clouding being close to BSA-PASG. If it was further raised from 4 to 5%, XR would not have been increased because X_min_ would have exceeded X_max_. Therefore, the NP14 concentration of 4% was selected in the proposed NP14-PASG. Further studies are required to optimize the concentration of NiS in order to increase XR while keeping X_min_ as low as possible.

NP14 was found to have an influence on the clouding temperature (_h_T_s_) and was sensitive to the optical transparency (X_min_) at room temperature. Figure 7 displays the optical opacity against temperatures on heating and cooling for 4 %(*w*/*v*) NP14 solution, 4 %(*w*/*v*) NP14-PAG, and 4 %(*w*/*v*) TM NP14 PASG. The 4 %(*w*/*v*) NP14 in water and PASG, the XR increased because X_max_ increased while X_min_ remained unchanged, whereas XR would not have increased in the NP14-PAG because X_min_ exceeded X_max_. This leads to the self-association of molecules into micelles, which are spherical or elongated structures. These aggregates form as a result of spontaneous self-aggregation at sufficiently high concentrations of amphiphilic, above a critical micelle concentration (CMC). While X_min_ in the NP14-PAG was similar to the NP14-solution, the hydrophilic component of polysaccharide is restricting the self-aggregation of micelles at room temperature.

The clouding temperature (_h_T_s_) was shifted toward lower temperatures in the gels (PAG and PASG) to form clouding earlier than the nominal clouding in water. This feature is due to the fact that the micelles (aggregation of micelles) in the NP14 solution grew much faster with temperature than in the gels. It was further shifted in NP14-PAG due to slower aggregation with temperature in viscous like NP14-PASG. At the same time, the X_max_ of NP14-PASG is much higher than that of the NP14 solution and NP14-PAG. This feature may be due to the stable micellar formation in viscous gels. Further studies are needed to understand the clear mechanism of the clouding of NP14 in the gels. Although the clouding point is altered relatively on a large scale through additives, its precise control is possible by employing methanol, ethanol, and i-propanol as effective clouding point boosters, while n-butanol is used to decrease the clouding point [30,31]. The addition of 1 %(*v*/*v*) n-butanol to NiS solution results in decreasing the clouding point by about 2 °C, whereas the same amount of methanol or ethanol increases the point by about 3 °C.

The acoustic properties are very close in both BSA-PASG and NP14-PASG, as a small amount of NP-14 (4 %(*w*/*v)*) may not have any significant effect on them (Figure 3). The proposed phantom was designed to have acoustic properties similar to those of liver tissue. The added polyacrylamide was found to play a critical role in them, which enables us to easily modify the recipe for ultrasonically mimicking tissues other than the liver. 

Figure 8 displays the acoustic parameters including attenuation, speed of sound, and B/A, measured as the amount of distilled water was reduced by the same volume of polysaccharide added without altering other recipes in Table 1.

The attenuation coefficients measured for frequency in the range of 1~5 MHz are shown in Figure 8a as the polysaccharide concentration was set at 10 %, 20 %, 30 %, 40 %, and 50 % in *w*/*v*. The attenuation coefficient increased linearly with frequency at all the concentrations as commonly seen in those of biological tissue. The rate of rising in the attenuation coefficient with respect to frequency increases with the concentration. The attenuation coefficient was almost the same as that of liver tissue at 1 MHz (a popular frequency for HIFU surgery) at a polysaccharide concentration of 40 %(*v*/*v*), however, the rate with respect to frequency (0.6~0.8) was smaller by a factor of 2 of tissue [13]. Note that the ultrasonic attenuation coefficients of the proposed 40 %(*v*/*v*) PASG were matched to those of soft tissues, 0.3~0.5 dB/cm at 1 MHz [32].

The speed of sound was observed to increase with polysaccharide concentration, from 1535 ± 10 to 1651 ± 7 m/s as the concentration increased from 0 to 50 %(*w*/*v*) as shown in Figure 8b. The density varied from 1049 ± 11 to 1067 ± 13 kg/m^3^ for the same concentration changes, and thus the acoustic impedance changed from 1.60 ± 0.02 to 1.70 ± 0.03 Mrayls. The speed of sound and acoustic impedance was found to be close to those of liver tissue (1575 ± 10 m/s, 1.68 Mrayls) at the 40 %(*w*/*v*) of polysaccharide [20,21].

The nonlinear parameter (B/A) was linearly increased with polysaccharide concentration as shown in Figure 8c The B/A ranged from 5.2 ± 0.2 to 6.8 ± 0.2 as the polysaccharide concentration increased from 0 to 50 %(*w*/*v*), Note that the measured value of B/A of 6.4 ± 0.3 of the PASG with 40 %(*w*/*v*) of polysaccharide is closed to that of liver tissue (~6.8 ± 0.2).

Changes in the optical transparency with the temperature of the NP14-PASG (Figure 2) were measured in a distilled degassed water bath. The repeatability of the temperature-grayscale curve of the PASG was tested with three repeatable measurements and is illustrated in Figure 9. Note that, in order to measure it, the water was heated using a hot plate (PC-420D, Corning^®^, Chelmsford Street, MA, USA). In this way, the fastest rate of temperature increase was 0.05 °C/s. This rate is extremely slow when compared to HIFU irradiation which is expected to raise tissue temperature by more than 30 °C in less than a second [33]. It is unknown how quickly the NiS responds to temperature changes. If the response time is slow, the size of the thermal lesions captured with a CCD camera will be underestimated. Although the response time is fast enough, the accuracy of the thermal lesion images is limited to the time resolution of the CCD camera. The CCD camera employed for the present study captures 30 frames per second.

## 5. Conclusions

A new cost-effective reusable NP14-PASG was proposed, which was designed to mimic the physical, acoustic, and thermal properties close to the liver and to visualize the thermal lesion expected to be produced in the liver by HIFU. The NP14 goes through a unique reversible clouding process and was employed as a thermo-sensitive agent of the UTMP. The experimental tests successfully verified that the thermal lesions formed with the proposed NP14 (40 %*(w*/*v)*) PASG are almost the same as those visualized in the existing BSA-PASG. The study claims that the NP14-PASG is expected to replace the existing expensive BSA-PASG for more effectively monitoring the thermal lesion by the therapeutic ultrasound where the thermal mechanism is important, in particular, for the cost-effective quality assurance of the HIFU devices for liver tumor surgery.

## Figures and Tables

**Figure 1 bioengineering-09-00786-f001:**
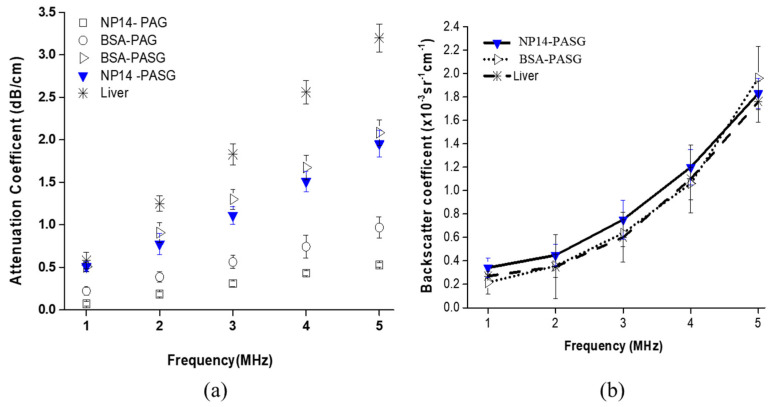
(**a**) Attenuation coefficients against frequency for the proposed NP-14 PASG ultrasonic tissue-mimicking phantom, together with those of BSA-PASG, BSA-PAG, and liver tissue [3,9,22] (**b**) Backscatter coefficients against frequency for the NP14-PASG, together with BSA-PASG and liver tissue [9,13]. Note that ultrasound does not scatter in the BSA-PAG where the backscatter coefficient cannot be measured [3]. The data points and the error bars represent, respectively, the mean and the standard deviation of five repeated measurements on different samples.

**Figure 2 bioengineering-09-00786-f002:**
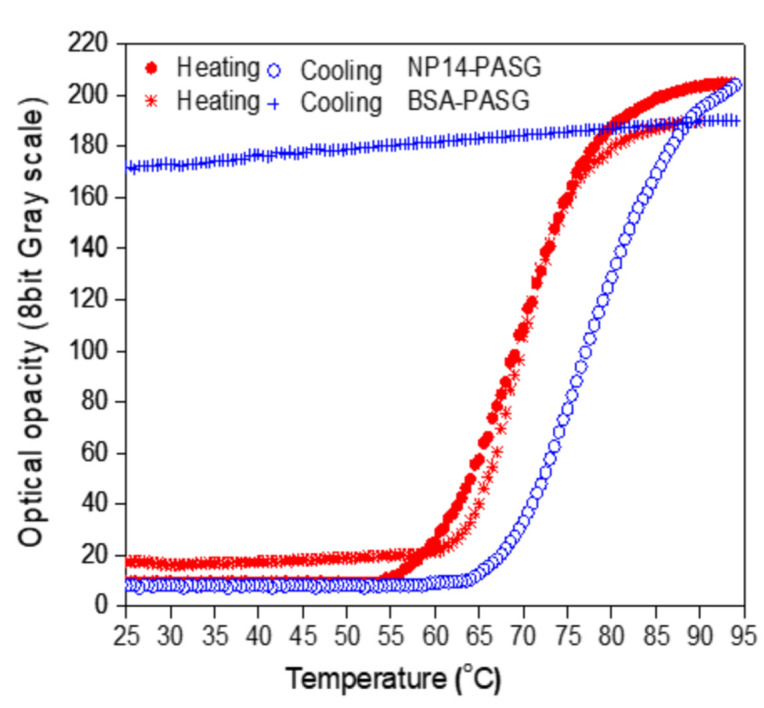
The relative optical opacity in terms of 8-bit grayscale (0~255) against temperature of the NP14-PASG measured when heating and cooling cycles from room temperature to 95 °C: NP14-PASG (● heating and ○ cooling) and BSA PASG (⁎ heating and  + cooling).

**Figure 3 bioengineering-09-00786-f003:**
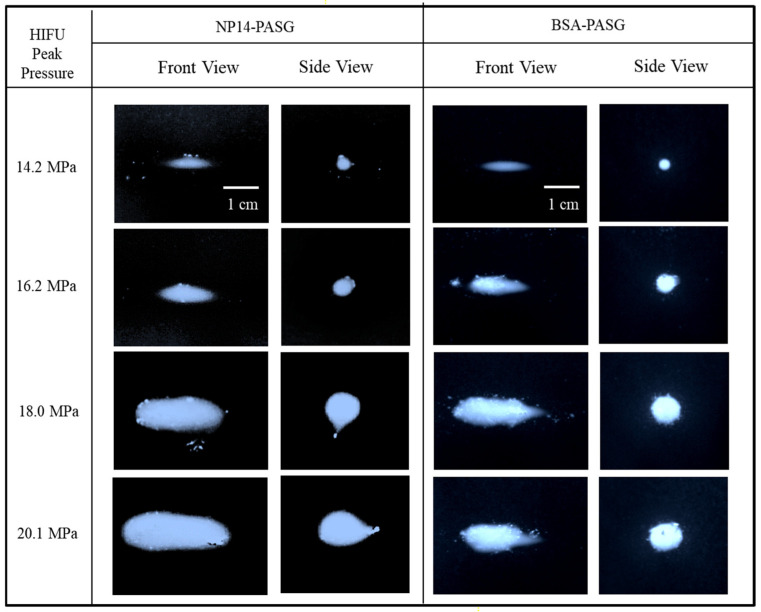
Typical images of the thermal lesions formed in the NP14-PASG (left two columns) in contrast to BSA-PASG (right two columns) exposed to the same HIFU fields with focal pressures (14.2 MPa, 16.2 MPa, 18.0 MPa, 20.1 MPa) for 10 s. In each phantom, the front views were shown in the left column and side view, in the right. The HIFU transducer operating in continuous wave mode at 1.1 MHz was placed at the left-hand side of the front view images. The increase in acoustic pressures causes accelerated lesion growth, distortion, and migration toward the HIFU transducer.

**Figure 4 bioengineering-09-00786-f004:**
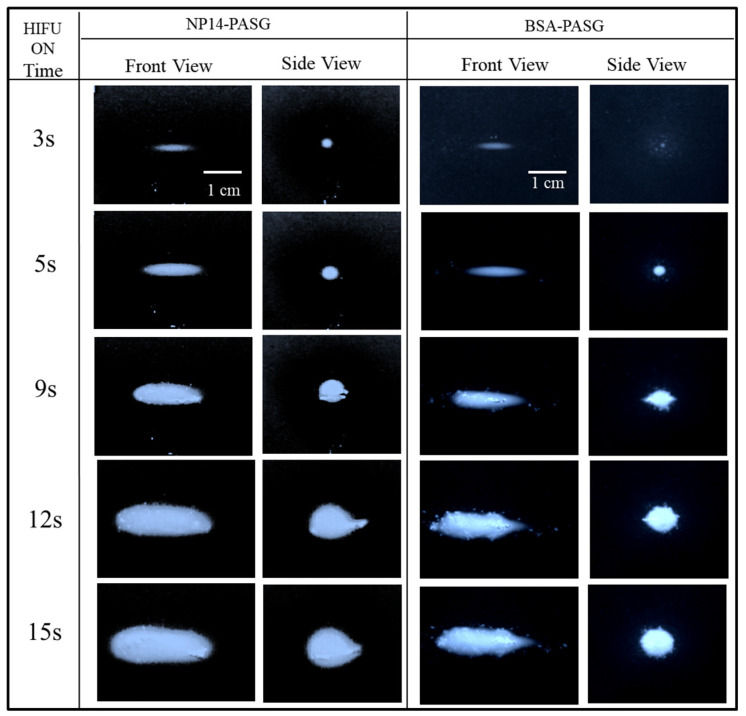
Typical images of the thermal lesions formed in the proposed NP14-PASG (left two columns) in contrast to BSA-PASG (right two columns) exposed to the same HIFU field for 3, 5, 9, 13, and 15 s. In each PAG, the front views were shown in the left column and side view on the right. The HIFU transducer was located on the left-hand side of the front view, operated at 1.1 MHz in continuous wave mode and producing a focal intensity of 4075 W/cm^2^.

**Figure 5 bioengineering-09-00786-f005:**
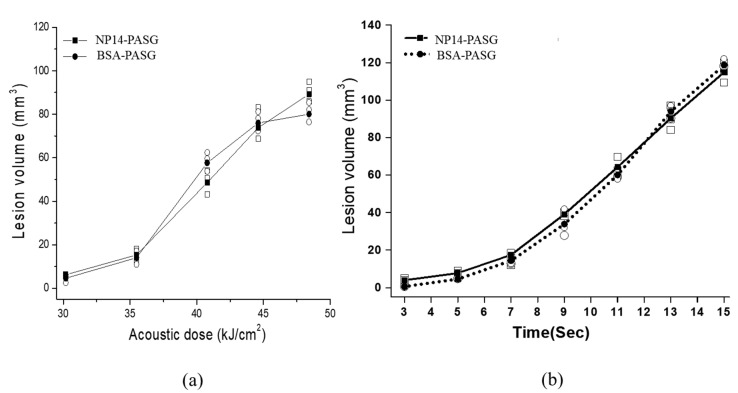
The thermal lesion volume against (**a**) the acoustic dose defined as the product of intensity and exposure time and (**b**) the HIFU exposure time under an intensity of 4080 W/cm^2^, measured on NP14-PASG (● mean, ○ data) and BSA-PASG (■ mean, □ data). Note that the mean volume was obtained with three repeated measurements on different samples.

**Figure 6 bioengineering-09-00786-f006:**
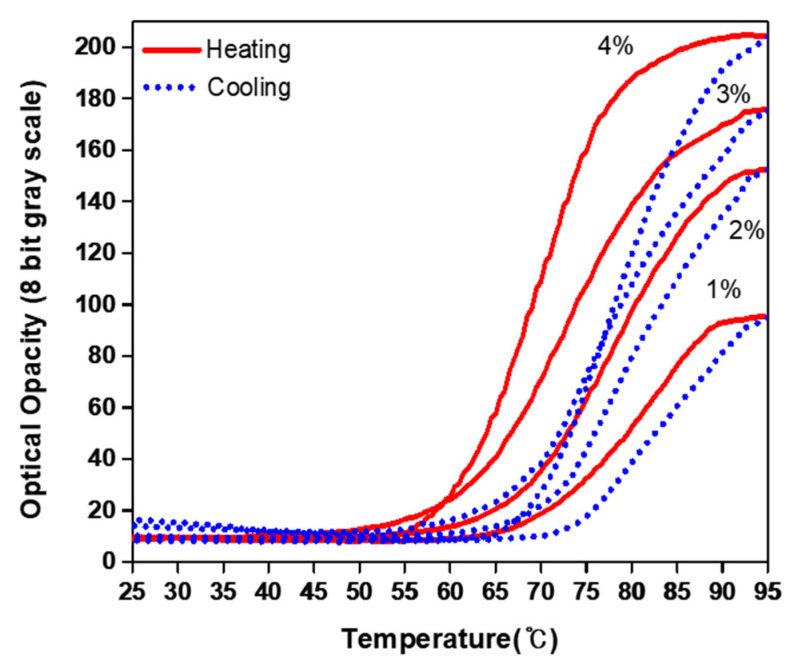
Reversible changes in the optical opacity (in 8-bit grayscale from 0 to 255) of the NP14-PASG against temperature, when varying the concentration of NP14 from 1% to 4 %(*w*/*v*): the solid line for heating and the dotted line for cooling.

**Figure 7 bioengineering-09-00786-f007:**
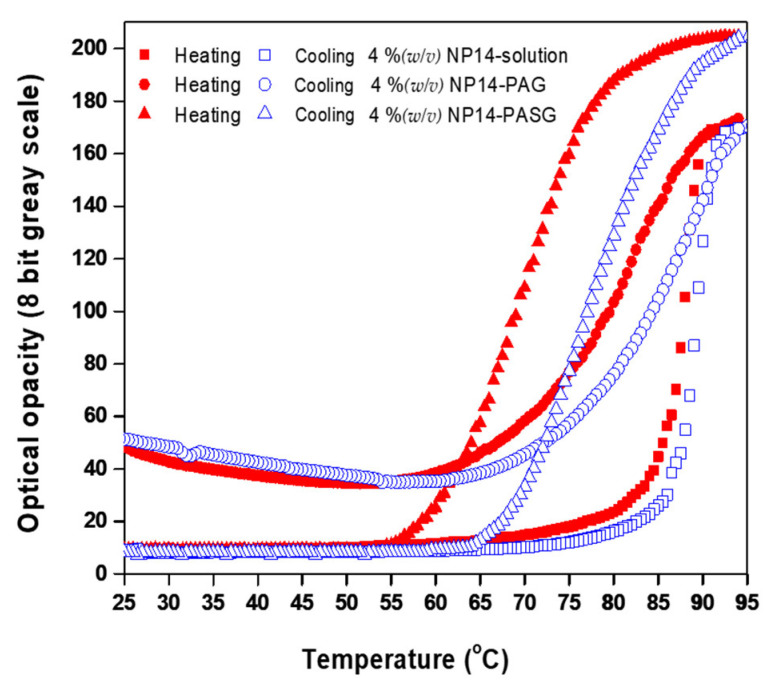
Reversible changes in the optical opacity (in 8-bit grayscale from 0 to 255) against temperature of the 4 %(*w*/*v*) NP14-PASG ( ▲ heating, ∆ cooling), in contrast to those of the 4 %(*w*/*v*) NP14-PAG (● heating, ○ cooling) and the 4 %(*w*/*v*) NP14.

**Figure 8 bioengineering-09-00786-f008:**
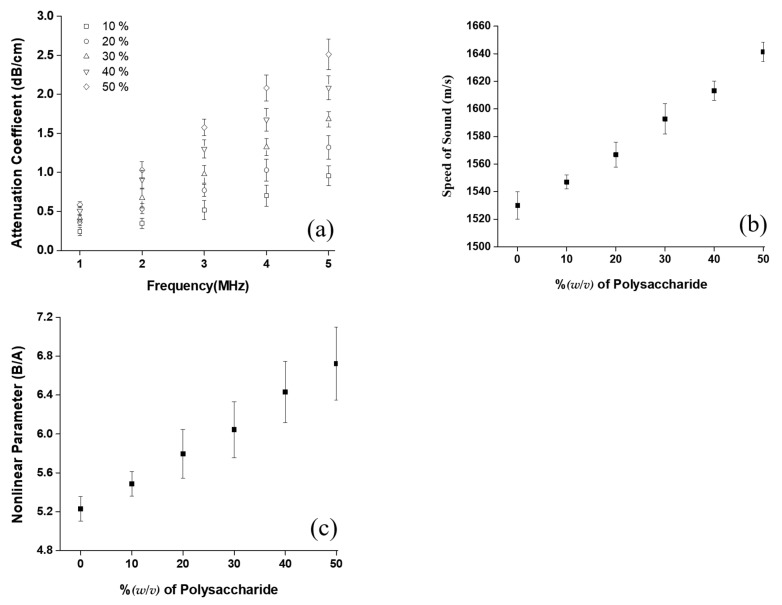
Variations in the physical properties of the PASG as the concentration of polysaccharide increases up to 50 %(*w*/*v*): (**a**) attenuation coefficients against frequency, (**b**) speed of sound, and (**c**) nonlinear parameter (B/A). The data points and the error bars represent, respectively, the mean and the standard deviation, which were obtained with five repeated measurements on different samples.

**Figure 9 bioengineering-09-00786-f009:**
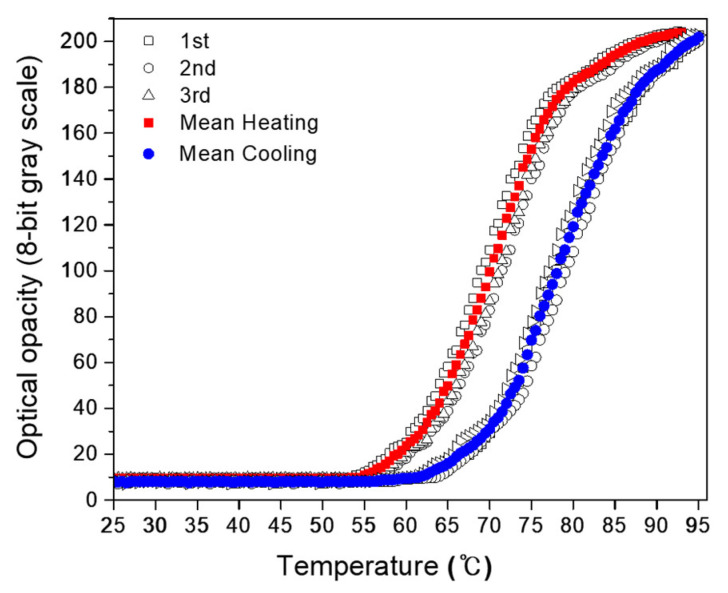
Reversible changes in the relative optical opacity (in 8-bit grayscale from 0 to 255) of the NP14-PASG versus temperature of heating and cooling cycles from room temperature to 95 °C. Data points represent three repeated measurements on different samples: heating (□ ○ △data, ■ mean) cooling (□ ○ △ data, ⬤ mean).

**Table 1 bioengineering-09-00786-t001:** Composition of the 50 mL polyacrylamide polysaccharide hydrogel (PASG) containing NP14 whose concentration is 4% in *w*/*v*.

Components	Quantity (mL, g)	Proportion (% in *v*/*v* or *w*/*v*)
Distilled Water	27.80 mL	55.60 %(*v*/*v*)
Polysaccharide *	20 g	40 %(*w*/*v*)
NP-14 **	2 g	4 %(*w*/*v*)
40 %(*w*/*v*) Acrylamide	8.75 mL	17.50 %(*v*/*v*)
Sodium Azid	0.1 g	0.20 %(*w*/*v*)
Potassium Iodide(KI)	0.5 g	1 %(*w*/*v*)
Glass Bead (40~80 µm)	0.01 × 10^−1^ g	0.02 × 10^−2^ %(*w*/*v*)
10 %(*w*/*v*) APS	0.250 mL	0.50 %(*v*/*v*)
TEMED	0.10 mL	0.20 %(*v*/*v*)

* Polysaccharide (corn syrup) is highly viscous liquid, so its volume could not be accurately measured using a micropipette. ** NP14 is supplied in solid states.

**Table 4 bioengineering-09-00786-t004:** Thermal properties of the NP14-PASG, compared with those of the BSA-PASG and liver tissue [9,16]. Measurements were carried out at room temperature (~23 °C) and three measurements were repeated on the different samples to obtain their mean and standard deviation. Note that the values in round brackets are the ratios of the mean values of the PASG to those of liver tissue.

Thermal Parameters	Unit	Liver	BSA PASG	NP14-PASG
Specific heat capacity	mJ m^−3^ C^−1^	3.628 ± 0.078	3.933 ± 0.028 (1.084)	3.910 ± 0.033 (1.078)
Thermal conductivity	W/m/°C	0.572 ± 0.009	0.562 ± 0.009 (0.983)	0.558 ± 0.01 (0.976)
Thermal resistivity	m/W/°C	1.748 ± 0.090	1.77 ± 0.027 (1.013)	1.79 ± 0.04 (1.024)
Thermal diffusivity	mm s^−2^	0.128 ± 0.030	0.151 ± 0.004 (1.180)	0.15 ± 0.003 (1.172)

**Table 5 bioengineering-09-00786-t005:** The characteristic parameters of an optical opacity against temperature: optical transparency range (T_s_), lesion contrast (T_e_), temperature range (TR) and slope (S) when clouding and clearing, measured on the NP14-PASG, in contrast to those of BSA-PASG [6,12] for the definitions of the parameters used and note that the unit gs represents the 8-bit grayscale from 0 to 255.

	Parameter	Unit	BSA-PASG	NP14-PASG
Heating (clouding)	_h_T_s_	°C	62	57
	_h_T_e_	°C	83	84
	_h_TR	°C	21	27
	_h_S	gs/°C	8.0	6.9
Cooling (clearing)	_c_T_s_	°C	90	87
	_c_T_e_	°C	-	66
	_c_TR	°C	-	21
	_c_S	gs/°C	-	8.4
Optical transparency	X_min_	gs	17	10
	X_max_	gs	190	203
	XR	gs	173	193
	C	%	72.7	78.8

**Table 6 bioengineering-09-00786-t006:** Characteristic parameters of an optical transparency against temperature: optical transparency range (_h_T_s_), lesion contrast (_h_T_e_), temperature range (_h_TR) and slope (_h_S) when clouding, measured on the NP14-PASG at a concentration of from 1 to 5 %*(w/v)*. Refer to Park et al., 2010 [6] for the definition of the parameters used and note that the unit gs represents the 8-bit gray scale of from 0 to 255.

	Parameter	Unit	1 %	2 %	3 %	4 %	5 %
Clouding (heating)	_h_T_s_	°C	67	63	61	57	54
	_h_T_e_	°C	90	88	87	84	82
	_h_TR	°C	22	25	26	27	28
	_h_S	gs/ °C	6.0	6.2	6.6	6.9	6.5
Optical transparency	X_min_	gs	3	5	8	10	12
	X_max_	gs	140	165	185	203	201
	XR	gs	137	160	177	193	189
	C	%	54.4	64.0	71.7	78.8	77.8

## Data Availability

Not applicable.

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
