# Peer review of "A Cost-Effective Reusable Tissue Mimicking Phantom for High Intensity Focused Ultrasonic Liver Surgery"

_bioengineering, 2022, doi:10.3390/bioengineering9120786_

Round 1

Reviewer 1 Report

The submitted paper concerns the fabrication and characterization of tissue-mimicking phantoms based on hydrogels for studies on HIFU-induced lesions. The proposed re-usable phantom material changes its properties not due to protein denaturation caused by high-temperature increase, but due to visible segregation of surfactant molecules added to the phantom body. The description of the experiments is rather clear. However, the minor revision of the manuscript is still required. The specific comments are attached below.

1. In Materials and Methods section (section 2.2), the description of acoustic measurements is provided. The Authors said that non-linearity coefficient (B/A) was measured by a broadband transducers (line 128-129), but in line 130, the Authors said that B/A coefficient was measured by a narrow-band transducers. It should be verified and rewrite to make it more clear for the readers.

2. The Authors frequently referred to their previous works from 2010, 2013 and 2014 when describing experimental setups and procedures. The question about the novelty of this paper is, then, reasonable. What was new in the current project that distinguished this from previous ones?

3. The Results section should be uniformly divided into subsection for the clarity. It is not clear why the beginning of Results are not affiliated into any subsection, but in line 258, the subsection 3.1 appears. The names of subsection should not include “:”.

4. What are the advantages of the presented method of determing post-HIFU lesions in phantoms (that has been reported many times in the literature) over the conventional thermometry or post-surgical evaluation of tissue injuries? For findamental research, it should not matter how invasive is a procedure to determine lesions. Can this method be used in real clinics? 

The language of the manuscript is informative and understandable. However, many linguistic and technical issues can be found in the text, as listed below:

a) The Authors should pay attention to the Journal’s style requirement.The references should be cited in [], the references to figures in the text of manuscript should be done as “Figure 1”, “Figure 2” (not like “Fig. 1” etc.). 

b) In line 29, “et al.” is not properly edited.

c) In lines 54-62, there is a problem with the font size and the presentation of special symbols. It must be corrected. The same concerns also other parts of the text (e.g., line 61, line 95, line 285, the caption of Table 5), therefore the careful proofreading is required.

d)  In line 68, the Authors mentioned “ultrasound scatters”, but probably they meant ultrasound scatterers (scattering agents).

e) In the sentence in line 83, the grammar issues occur.

f) The sentence in line 99 should not be continued after the Table 1.

g)The MDPI style requires the addition of all details on the providers of chemicals and equipment. For instance, in line 108, the city of producer is missing as well as in line 136.

h) Generally, used abbreviations should be explained even though are commonly used, for instance, in line 145, CCD camera.

i) The unit of density in line 194 should be verified (m3 or cm3?)

j) The sentence in line 210 is merely clear. Moreover, in line 212, “a bit” more seems to be not a physical term.

Reviewer 2 Report

abstract needs to be rewritten via adding some numeric data 

there are many technical as well as grammar mistakes need to be solved

there is no comparison between current study with published work 

there are many articles related to the cost-effective reusable tissue-mimicking phantom kindly make answer

results are two generic need to explore the results context with their impact on the degradability factor

conclusion section should be enhanced via adding some future recommendations related to further research work on cost-effective reusable tissue-mimicking phantom

Reviewer 3 Report

The study was to propose a reusable cost-effective UTMP made of a polyacrylamide polysaccharide hydrogel containing NP14 of an NiS of the Polyoxyethylene Nonylphenyl Ethers series, as a temperature-sensitive agent. Overall, this study is well-designed and the manuscript is well-written. I just have a minor concern.

In the title and introduction, liver was mentioned. However, I cannot see the association with liver in the abstract and the resuls section. This issue needs to be clarified.
